# Non-Invasive Biomarkers for Celiac Disease

**DOI:** 10.3390/jcm8060885

**Published:** 2019-06-21

**Authors:** Alka Singh, Atreyi Pramanik, Pragyan Acharya, Govind K. Makharia

**Affiliations:** 1Department of Gastroenterology and Human Nutrition; All India Institute of Medical Sciences, New Delhi-110029, India; singhalka34@gmail.com; 2Department of Biochemistry, All India Institute of Medical Sciences, New Delhi-110029, India; atreyipram91@gmail.com (A.P.); dr.pragyan.acharya@gmail.com (P.A.)

**Keywords:** celiac disease, biomarker, serology, enteropathy

## Abstract

Once thought to be uncommon, celiac disease has now become a common disease globally. While avoidance of the gluten-containing diet is the only effective treatment so far, many new targets are being explored for the development of new drugs for its treatment. The endpoints of therapy include not only reversal of symptoms, normalization of immunological abnormalities and healing of mucosa, but also maintenance of remission of the disease by strict adherence of the gluten-free diet (GFD). There is no single gold standard test for the diagnosis of celiac disease and the diagnosis is based on the presence of a combination of characteristics including the presence of a celiac-specific antibody (anti-tissue transglutaminase antibody, anti-endomysial antibody or anti-deamidated gliadin peptide antibody) and demonstration of villous abnormalities. While the demonstration of enteropathy is an important criterion for a definite diagnosis of celiac disease, it requires endoscopic examination which is perceived as an invasive procedure. The capability of prediction of enteropathy by the presence of the high titer of anti-tissue transglutaminase antibody led to an option of making a diagnosis even without obtaining mucosal biopsies. While present day diagnostic tests are great, they, however, have certain limitations. Therefore, there is a need for biomarkers for screening of patients, prediction of enteropathy, and monitoring of patients for adherence of the gluten-free diet. Efforts are now being made to explore various biomarkers which reflect different changes that occur in the intestinal mucosa using modern day tools including transcriptomics, proteomics, and metabolomics. In the present review, we have discussed comprehensively the pros and cons of available biomarkers and also summarized the current status of emerging biomarkers for the screening, diagnosis, and monitoring of celiac disease.

## 1. Introduction

Celiac disease (CeD) is a systemic autoimmune disorder that is induced by the ingestion of gluten protein present in wheat, barley, and rye in genetically predisposed individuals [1]. Once considered to be limited to Western Europe, CeD has now emerged as a major public health problem globally. A recent systematic review suggests that 0.7% (95% Confidence Interval (CI), 0.5–0.9%) of the global population suffers from CeD [2]. With the global population of 6.4 billion, approximately 37–59 million people are estimated to have CeD globally. While the global pool of patients is so large, the majority of patients (83–95%) in developed countries, and possibly even a higher number in developing countries, still remain undiagnosed [3]. This large pool remains unrecognized partly because of the lack of classical gastrointestinal symptoms in approximately half of patients. The spectrum of clinical manifestation of CeD is wide and includes both gastrointestinal symptoms such as chronic diarrhea, dyspepsia, anemia, and failure to thrive and extra-gastrointestinal manifestations such as short stature, dermatitis herpetiformis, infertility, and liver diseases. Recognition of wide spectrum of CeD, simplification of the diagnostic criteria and widespread use of celiac-specific serological tests (anti-tissue transglutaminase antibody, anti-endomysial antibody or anti-deamidated gliadin peptide antibody) have led to an increase in the recognition of CeD globally [4]. There is no single gold standard test for the diagnosis of CeD, and the diagnosis of CeD is based on a combination of clinical manifestations, presence of the celiac-specific serological test and demonstration of villous abnormality on intestinal mucosal biopsies [5]. While the treatment of CeD at present time is life-long gluten-free diet (GFD); a number of drugs including intraluminal gluten degrading enzyme therapies such as latiglutenases, peptide vaccines such as NexVax 2, and zonulin antagonist such as larazotide are being explored as an alternative or additive treatment for patients with CeD.

A biomarker is a defining characteristic that is measured as an indicator of normal biological processes, pathogenic processes, or responses to an exposure or intervention, including therapeutic interventions [6]. Since an ideal biomarker should provide a “signature” for a condition, the development of biomarkers requires several rounds of controlled experimentations, and validation before it can be used in clinical practice. In the present review, we have discussed comprehensively the pros and cons of available biomarkers and we have also summarized the current status of emerging biomarkers for the screening, diagnosis, and monitoring of the disease. For CeD, biomarkers are required for multiple purposes including a) screening of the disease, b) detection/prediction of enteropathy, c) markers of complications, d) monitoring of the disease, and e) assessment of adherence to gluten free diet (GFD). Indeed, enormous efforts have been made in this direction; some of the established and potential biomarkers are summarized in Table 1.

## 2. Serological Biomarkers

Serological tests are the first line investigation for the screening of patients suspected to have CeD. Until the late 1950s, there was no biomarker for the diagnosis of CeD, and the diagnosis made was on the basis of the presence of suggestive clinical symptoms and resolution of symptoms with a GFD [7]. The advent of methods for obtaining intestinal biopsies such as Crosby capsules and endoscopic techniques, added villous atrophy as one of the most specific requirements for the diagnosis of CeD. The discovery of anti-gliadin antibodies (AGA) in the 1960s was a landmark step in the evolution of the modern-day diagnostic strategy for CeD. AGA remained the first line celiac-specific serological test until the 1990s [8]. In the 1990s, anti-endomysial antibody (AEA) was discovered and a combination of AGA with AEA testing became the standard diagnostic strategy for CeD [9]. With recognition of the high false positive rate for AGA, the use of AGA fell both for the screening and diagnosis of CeD. Further discovery of anti-tissue transglutaminase (tTG) as the substrate for AEA, tTG based enzyme-linked immunoassays (ELISA) became the standard diagnostic test for CeD [10].

### 2.1. Anti-Gliadin Antibodies

Anti-Gliadin Antibodies (AGA) is produced against gliadin, a prolamin found in wheat and related cereals. Anti-Gliadin Antibodies are of two types IgA-AGA and IgG-AGA. Anti-Gliadin Antibodies is no more used for the diagnosis of CeD because of the advent of more reliable serological tests. However, there has been renewed interest in the utility of AGA. Immunoglobulin G (IgG) AGA and IgA-AGA are now used to recognize other gluten-related disorders such as non-celiac gluten sensitivity, gluten ataxia, and autism [11]. Anti-Gliadin Antibodies (more precisely, IgG dependent AGA) is positive in approximately 50% of patients with non-celiac gluten sensitivity [12,13].

### 2.2. Anti-Endomysial Antibody

Chorzelski et al. discovered the AEA test, which revolutionized the diagnostic strategy for CeD [14]. Anti-Endomysial Antibody (AEA) is an antibody against the smooth muscle’s inter-myofibrillar substance [9]. Earlier, a monkey esophagus was used as a substrate for AEA testing and the use of human umbilical cord as a substrate has enhanced the sensitivity and specificity of AEA [15]. While both sensitivity and specificity of AEA was reported to be >95%, a recent systematic review has reported a lower pooled sensitivity, 73.0% (95% CI, 61.0% to 83.0%), based on the data of newer studies, the specificity however still remains to be 99.0% (95% CI, 98.0% to 99.0%) [16]. While AEA is very specific, detection of AEA requires indirect immunofluorescence, which is labor intensive and time -consuming as compared to the estimation of tTG Ab by ELISA [17].

### 2.3. Anti-Tissues Transglutaminase Antibody

Transglutaminase (TG) is a calcium-dependent enzyme, which catalyzes the covalent bond and cross-linking of proteins irreversibly [18]. Nine different types of the TG gene have been discovered in mammals, eight codes for catalytically active enzymes and one for an inactive enzyme. These TGs play different roles in different tissues in physiological and pathological conditions. Transglutaminase 1 (TG1) (keratinocyte), TG3 (epidermal), and TG5 are involved in the formation of the cornified envelope during keratinocyte differentiation, thus contribute to the cutaneous barrier function [19]. Transglutaminase 6 (TG6) and TG7 are expressed in testis, lung and brain, but their function is still uncertain [20]. Transglutaminase 2 (TG2) is ubiquitously present in cells and tissues—and hence TG2 is known as “tissue” TG.

Autoantigen against TG2 was identified by Dieterich et al., who also suggested the role of TG2 in the deamidation of the bond between glutamine and lysine, present in gluten [18]. Of all the serological tests, IgA anti-TG2 Ab is the most widely used test both for the diagnosis and initial screening for CeD because of its very high sensitivity and specificity, ease of use, and its quantitative capability. In a recent systematic review, Chou R et al. reported a pooled sensitivity of anti-tTG Ab to be 92.8% (95% CI, 90.3–94.8%); specificity 97.9% (95% CI, 96.4–98.8%); a positive likelihood ratio (LR) of 45.1 (95% CI, 25.1–75.5%) and negative LR of 0.07 (95% CI, 0.05–0.10%) [16]. Immunoglobulin A (IgA) tTG levels also correlate with the degree of severity of mucosal damage, and a titer of 10 folds or higher over the upper limit of normal (ULN) predicts presence of villous abnormality with very high specificity [21].

Antibodies against other TGs have been reported in extra-intestinal forms of gluten-related disorders [22]. Anti-TG3 Ab has been reported in patients having dermatitis herpetiformis [23]. As discussed above, TG6 is found in the brain and anti-TG6 Ab has been reported in gluten-induced neurological diseases such as gluten ataxia [24,25]. The concentration of anti-TG6 Ab has been observed to correlate with longer gluten exposure in them and the level of TG6 decreases after GFD [26]. In a study, 73% of patients with idiopathic sporadic ataxia positive for AGA, were also positive for TG6 antibodies [27].

### 2.4. Anti-Deamidated Gliadin Peptides

Anti-deamidated gliadin peptides Ab (DGP) is directed against deamidated gliadin peptides and is another serologic marker for the diagnosis of CeD [28]. Initially, IgA DGP was reported to be equally sensitive and specific as IgA tTG Ab, however recent studies have shown that tTG Ab is the most trusted serological test for CeD [29,30]. A recent systematic review and meta-analysis reported the pooled sensitivity of anti-DGP Ab to be 87.8% (95% CI, 85.6–89.9%), and of specificity 94.1% (95% CI, 92.5–95.5%) [16].

### 2.5. Point-of-Care Test

Point-of-care tests (POCTs) for the diagnosis and monitoring of CeD have been in use for the past one decade, especially in Europe [31,32]. They are easy to perform, do not require a laboratory or experienced laboratory staff, and have a quick turn-around time [33,34]. Therefore, POCTs have the potential to increase CeD diagnosis rates worldwide, facilitate early diagnosis, and reduce cost. POCTs have been shown to be successfully used in various settings including primary care, specialty clinics, and endoscopy suites [35]. The majority of these POCTs are immunochromatographic tests and they are performed in a similar way with whole blood/serum and buffer placed on a test field that diffuses down a test strip [34]. If antibodies (tTG and/or DGP) are present, antigen-antibody complexes are detected by labeled anti-human IgA and/or IgG antibodies. The test is visually read on site after a few minutes as recommended by the manufacturer. A positive test is reflected by the presence of a solid line in the test window and a negative test by the absence of a line in the test window [31,32,33,34,35].

In a recent systematic review and meta-analysis, we observed the pooled sensitivity and specificity of all POCTs (based on tTG or DGP or tTG + anti-gliadin antibodies) for diagnosing CeD to be 94.0% (95% CI, 89.9–96.5%) and 94.4% (95% CI, 90.9–96.5%), respectively. The pooled positive and negative LR for POCTs were 16.7 and 0.06, respectively [36].

### 2.6. Limitations of Serological Tests

False positive results can occur due to a cross-reaction of antibodies in conditions such as enteric infection, chronic liver disease, congestive heart failure, or hypergammaglobulinemia [37]. The serological tests should ideally be conducted when the patient is on a gluten-containing diet, as being on a low-gluten diet or gluten-free diet can lead to a false negative result [38]. False negative results may also be due to IgA deficiency, which affects 2–3% of the general population [39]. In IgA deficient patients, an IgG -based test such as IgG DGP or IgG anti-tTG antibodies should be performed [12].

A few studies have raised question about the sensitivity of these assays in clinical practice [40,41]. tTG antigens used in these commercial kits are variable, ranging from recombinant human tTG to human tTG cross-linked to gliadin specific peptides [42,43]. In addition, commercial kits typically provide sensitivity and specificity values that are calculated using small, poorly-defined populations, which can be misleading. Several studies comparing different anti-tTG-ab based assays from different manufacturers have revealed variable sensitivities and specificities for detecting CeD; however, most of these studies were small in size and did not have the necessary sample size to accurately comment on the diagnostic accuracies of the testing [22,43,44,45,46]. In addition to the possibility of inter-assay variation in the diagnostic performance of commercially available IgA anti-tTG-ab assays, there might be intra-assay variation in the diagnostic performance of these assays for different ethnic populations. Studies from India suggest that the sensitivity of several tTG-ab ELISA assays might be lower in the Indian population than that reported in the Caucasian population [33,36].

## 3. Genetic Markers

### Human Leukocyte Antigen (HLA) DQ Haplotyping

Celiac disease is strongly associated with Human Leukocyte Antigen (HLA) and approximately 95% of CeD patients express HLA-DQ2 encoded by DQA1*05 and DQB1*02 and the rest 5% carry DQ8 alleles encoded by DQA1*03 and DQB1*0302 alleles [47]. The HUGO Gene Nomenclature Committee (http://www.genenames.org/) has shown HLA-DQA1 and HLA-DQB1 class II genes as CELIAC1 [48]. HLA alone accounts for about 40% of genetic heritability for CeD, while 60% genetic susceptibility of CeD by non-HLA genes [49,50]. Other that HLA -DQ2/ DQ8, more than 40 candidate genes have been discovered to be associated with CeD [51,52]. Interestingly, approximately 30% of the general population have HLA-DQ2/DQ8 haplotype, only 3% of them ever develop gluten-related disorders [53].

Human leukocyte antigen typing is not sufficient for the diagnosis of CeD because of its modest sensitivity (HLA-DQ2, 70–99.8%; HLA-DQ8, 1.6–38%) and specificity (HLA-DQ2, 69–77%; HLA-DQ8, 77–85%) [54]. Nevertheless, HLA-DQ typing test has a high negative predictive value that suggests if an individual is negative for HLA-DQ typing, he or she is less likely to have CeD [49]. Human leukocyte antigen (HLA)-DQ typing is mainly used for the exclusion of CeD and it is considered to be an additional test especially in patients where no agreement exists between the serological and histological results.

## 4. Biomarkers to Predict Presence of Enteropathy

Villous atrophy is the hallmark of CeD, which gets reverted after the institution of GFD. While the clinical response to GFD is evident in weeks and months; reversal of mucosal changes takes months and even years [2,54]. Furthermore, despite GFD, complete villous recovery may not be achieved. The small intestinal mucosal biopsy is the cornerstone for the diagnosis of CeD. In addition to being the gold standard for the initial diagnosis of CeD, periodic biopsies are also recommended for the monitoring of the disease. However, obtaining biopsies is an invasive and expensive procedure (endoscopic examination). Moreover, a correct assessment of biopsies requires an experienced pathologist and well-oriented high-quality biopsy specimens. In fact, an active debate is going on amongst the celiac disease scientific community, whether to do biopsies or skip biopsies in the making of a CeD diagnosis. A relevant question is “can we demonstrate/predict villous atrophy by non-invasive means?”

Enterocytes are very specialized cells and they perform specific functions including absorption of nutrients and secretion of enzymes. Because of the constant exposure of the gastrointestinal (GI) tract to harsh mechanical and chemical conditions, the GI tract has evolved mechanisms to cope with these assaults via a highly regulated process of self-renewal [55]. Most of the epithelial cells are replaced every 3 to 5 days. According to the so-called “Unitarian hypothesis”, first proposed by Cheng and Leblond, epithelial cell renewal is driven by a common intestinal stem cell residing within the crypt base [56,57]. From their niche, intestinal stem cells (ISCs) give rise to transit-amplifying cells that migrate upwards and progressively lose their proliferative capability and maturate to become fully-differentiated villous epithelial cells (absorptive enterocytes or secretory cells which include goblet cells, enteroendocrine cells, paneth cells, and tuft cells). Each adult crypt harbors approximately 5 to 15 ISCs that are responsible for the daily production of about 300 cells; up to 10 crypts are necessary to replenish the epithelium of a single villus [58]. According to mathematical modeling, approximately 1400 mature enterocytes are shed from a single villus tip in 24 h (2 × 10^8^ cells shed from small intestine every day) [58].

All the changes that occur in the intestinal mucosa include intraepithelial lymphocytosis, heightened apoptosis of enterocytes, heightened regeneration of enterocytes, imbalance in the rate of apoptosis and regeneration leading to decrease in the villous height, enterocyte mass, and increase in the inflammatory cells in the mucosa [59,60].

There are certain molecules which can serve as a biomarker of the enterocyte mass and enterocyte function (citrulline, cytochrome P450 3A4), enterocyte injury (intestinal fatty acid binding protein), and enterocyte regeneration (regenerating gene 1α).

### 4.1. Cytochrome P450 3A4

Like the liver, intestinal mucosa also has a drug metabolizing enzyme system along with the crypt- villous axis. Cytochrome P450 3A4 is a drug-metabolizing enzyme system expressed abundantly at the tips of the villi, and less abundantly at the crypts [61]. Loss of intestinal villi because of any cause including CeD can lead to a reduction in the activity of CYP3A4. Both the expression and function of CYP3A4 can be assessed to estimate the function of enterocytes. Simvastatin is a lipid- lowering agent that is metabolized by CYP3A4. Therefore, the function of CYP3A4 can be assessed by pharmacokinetics and maximum concentration (C_max_) of orally administered simvastatin (SV) in blood. While healthy people having normal enterocyte function should have a low level of SV and higher levels of its metabolites in the blood; the levels of SV should be high and its metabolites level should be low in those having enteropathy. Therefore, the assessment of the functional activity of CYP3A4 may serve as a biomarker for enteropathy [62,63].

As a preclinical test of this hypothesis, we measured the plasma concentrations of SV and its major metabolites in mice expressing the human CYP3A4 transgene in the small intestine, in whom acute enteropathy had been induced using polyinosinic-polycytidylic acid (poly I:C). In CYP3A4-humanized mice, a marked decrease in simvastatin metabolism was observed in response to enteropathy. Encouraging results from these experiments motivated us to do a clinical study involving untreated as well as treated patients with CeD along with healthy controls, in order to determine the potential utility of using serum concentrations of SV and/or its metabolites as a tool predicting enteropathy. We included 11 healthy volunteers, 18 newly diagnosed patients with CeD, and 25 patients with CeD who had followed a GFD for more than one year. The C_max_ of orally administered SV, plus its major non-CYP3A4 derived metabolite SV acid (SV_eq_ C_max_) was measured, and compared to clinical, histological, and serological parameters. Untreated patients with CeD displayed a significantly higher SV_eq_ C_max_ (46 ± 24 nM) compared to treated patients (21 ± 16 nM, *p* < 0.001) or healthy subjects (19 ± 11 nM, *p* < 0.005). SV_eq_ C_max_ correctly predicted the diagnosis in 16/18 untreated celiac patients as well as the recovery status of all follow-up patients. Therefore, SV_eq_ C_max_ is a promising non-invasive marker for the assessment of small intestinal health. Further studies are warranted to establish its clinical utility for assessing the status of villous abnormality in patients with CeD.

### 4.2. Plasma Citrulline

Citrulline is a non-protein amino acid that is mainly synthesized by the enterocyte and hence the level of citrulline in plasma can represent the synthetic function of the enterocytes [64]. The concept of using citrulline as a marker of enterocyte mass was first provided by Crenn et al. [65]. They reported lower citrulline levels in the plasma of patients with short bowel syndrome compared to controls (20 ± 13 vs. 40 ± 10 μmol L^−1^, *p* < 0.001). Furthermore, the level of plasma citrulline also correlated with the length of the residual intestine [66]. A lower level of plasma citrulline in comparison to healthy controls has been observed in many small intestinal diseases including CeD, giardiasis, tropical sprue, and small bowel lymphoma. Interestingly, a declining trend in the levels of plasma citrulline was observed with increasing severity of villous atrophy i.e., concentration of citrulline <10 μmol L^−1^ in patients with diffuse total villous atrophy, 10–20 μmol L^−1^ in patients with proximal-only total villous atrophy, and 20–30 μmol L^−1^ for patients with partial villous atrophy. At an optimum cut-off value of plasma citrulline of 20 μmol L^−1^, the diagnostic accuracy of predicting villous abnormality was 92% with sensitivity and specificity of 95% and 90%, respectively [67].

In a recent study, we have shown that plasma citrulline of <30 µmol L^−1^ indicates the presence of villous abnormality of modified Marsh grade more than 2 with a diagnostic accuracy of 89% with sensitivity and specificity of 78.6% and 95.5%, respectively. Above mentioned statistics suggest that it is possible to predict significant villous abnormality based on the citrulline level even without obtaining duodenal mucosal biopsies in 78.6% patients with 95.5% specificity in patients suspected to have CeD. Therefore, in a clinical context of a positive anti-tTG Ab, if the plasma citrulline is less than 30 µmol L^−1^, one can predict that there is significant villous atrophy and one can choose to avoid duodenal biopsy for demonstration of villous atrophy. (Under publication) A recent meta-analysis by Fragkos et al. has shown that citrulline levels correlate with small bowel length in patients with short bowel syndrome (r = 0.76) while negatively correlate with the severity of intestinal disease such as CeD, tropical enteropathy, Crohn’s disease, mucositis, acute rejection in intestinal transplantation [67].

### 4.3. Intestinal-Fatty Acid Binding Proteins

Fatty acid-binding proteins (FABPs) are small (14–15 KDa) cytoplasmic proteins involved in the cholesterol and phospholipid metabolism, transport of long-chain fatty acids and maintenance of lipid homeostasis [68]. Fatty acid-binding proteins was first discovered in 1972 and by now nine types of FABP have been identified from the different organs where they are involved in active lipid metabolism. Intestinal type (I-FABP) is specifically expressed in intestine and encoded by the FABP2 gene present on chromosome 4 [69]. Intestinal fatty acid-binding proteins is expressed throughout the intestine, most abundantly in the jejunum, and in greater abundance in enterocytes at the villous tip than in the enterocytes at the crypts.

As I-FABP is expressed in enterocytes, the injury to enterocytes leads to the release of I-FABP at the local sites, which are then absorbed and passes into the circulation. As the levels of I-FABP get elevated in the serum in patients with enterocyte damage, I-FABPs has been considered to be a potential biomarker for the detection of enteropathy [70]. Elevated levels of I-FABP are detected in patients with necrotizing enterocolitis in preterm infants, mesenteric infarction, and intestinal allograft rejection [71,72,73,74,75].

In a study including 96 biopsy-proven adult CeD patients, I-FABP levels were higher in untreated CeD compared with controls (median 691 pg mL^−1^ vs. 178 pg mL^−1^, *p* < 0.001) and the level declined with GFD [76]. Interestingly, in those patients with CeD, where the FABP levels remained elevated, the biopsy showed persistent histological abnormalities. In another multicenter study, high FABP levels have been shown to predict the diagnosis of CeD in 61/90 (67.8%) children [77].

In a study of treatment naïve patients with CeD, we observed that the optimal cut-off value of plasma I-FABP is ≥1100 pg ml^−1^ in predicting villous abnormalities of modified Marsh grade 2 or more. Using the ROC curve analysis, we observed a diagnostic accuracy of 78% with sensitivity, specificity and odds ratio, LR+ and LR- of 39.7%, 95.5%, 13.9 (95% CI 5.8–33.1%), 8.7 and 0.63, respectively in predicting villous abnormality of modified Marsh grade 2 or more (under publication). We believe that I-FABP may be a good biomarker of enterocyte damage but it requires more studies around the world to understand its consistent performance. There are inconsistencies in the performance of I-FABP as a diagnostic marker as reported in earlier studies [74,76,78].

### 4.4. Regenerating Gene1α

Human *Reg1α* is a member of the multigene family and it plays a role in the regeneration of cells [79]. There are four types of the Reg gene, Reg1α is one of them which is also known as lithostathine-1-alpha or islet cell regeneration factor (ICRF) or islet of Langerhans regenerating protein (Reg) [80]. In the GI tract, the highest levels of expression of Reg1α have been observed in the small intestine and it is considered as a regulator of cell growth that is required to generate and maintain the villous structure [81]. A high level of the Reg1α may denote the effort of the small intestinal mucosa trying to compensate for the accelerated enterocyte injury/apoptosis/necrosis [82]. In patients with CeD where there is excessive apoptosis of enterocytes, there is a higher expression of the Reg1α gene in the intestinal crypts and the expression falls with GFD [83]. There is only one study showing elevated levels of Reg1α in patients with CeD (n = 40) compared to healthy controls (n = 35) and the levels declined after GFD [84]. There is a need for further study to prove whether Reg1α is a stable marker or not.

## 5. High-Throughput Technologies for Biomarker Discovery

Several studies have employed high throughput genomics, transcriptomics, and proteomics approaches to find out a panel of biomarkers to reveal the extent and status of small intestinal damage by CeD.

### 5.1. Transcriptomics Approaches

Since transcriptomics requires a lower amount of biological material such as intestinal mucosal biopsies, this approach has been used to explore various aspects of small intestinal biology including phenotypes and functions of T cells, as well as cytokine profiles within the small intestinal tissue [85]. In order to define the global cytokine gene expression network associated with CeD, one study demonstrated higher expression of interleukin (IL)-15, IL-18, and IL-21 with gluten ingestion, which could drive the inflammatory response in them [86]. In another study, a significant upregulation of 25 odd defense-related genes was observed, including IRF1, SPINK4, ITLN1, OAS2, CIITA, HLA-DMB, HLA-DOB, PSMB9, TAP1, BTN3A1, and CX3CL1 in intestinal epithelial cells of patients having active CeD in comparison with those treated with GFD [87]. Galatola M, et al. have explored the possibility of the development of a non-invasive biomarker obtainable from peripheral circulation by carrying out gene expression analysis using peripheral blood mono-nuclear cells (PBMCs) and they defined a 4 gene PBMC signature including NFKB, IL-21, LPP, and RGS1 for discriminating patients with CeD from non-CeD individuals [88]. Although these genes were significantly differentially expressed in the intestinal mucosa, their expression differences were much weaker in PBMCs. Subsequently the same group demonstrated a set of 9 genes including KIAA, TAGAP (T-cell Activation GTPase Activating Protein), and SH2B3 (SH2B Adaptor Protein 3), RGS1 (Regulator of G-protein signaling 1), TAGAP, TNFSF14 (Tumor Necrosis Factor Superfamily member 14), and SH2B3 which could differentiate patients with CeD from controls [89].

Two recent transcriptomics studies have made strides towards defining transcriptomics signatures in CeD. In one of the studies, we analyzed the transcriptomes in the intestinal mucosa of HLA DQ2/DQ positive asymptomatic first-degree relatives (FDRs) of patients with CeD and showed that pre-symptomatic FDRs harbored a transcriptomic signature that was distinct from controls in spite of the fact that these FDRs had no symptoms or enteropathy at the time of the study. This clearly suggests that there are phenotypic differences in individuals without active enteropathy, which may be exploited in order to develop a biomarker to predict intestinal damage. The second study demonstrated a CeD / no CeD diagnosis based on transcriptomic profiles of duodenal biopsies which revealed a potential biomarker subset consisting of CXCL10, GBP5, IFI27, IFNG, and UBD [90].

### 5.2. Proteomics Approach

Proteomics studies typically require a higher amount of biological material for studying the proteome profile in tissues. Recent advances in label-free quantitative approaches have enabled whole tissue proteomics of many tissues including small intestinal biopsies in patients with CeD [91]. In a recent study, biopsy specimens collected from patients with CeD before and after 1-year treatment with GFD showed differential expression of proteins such as Ig variable region IGHV5-51, which could serve as a specific marker of immune activation in patients with CeD.

Proteomics approaches have also been used to identify autoantigens in patients with CeD. Stulík J et al. used sera and intestinal biopsies from the patient with CeD and carried out 2D gel electrophoresis of the intestinal proteome followed by immunoblotting of matched patient sera in order to identify the repertoire of self-antigen in CeD patients [92]. This study detected 11 new self-antigen including Adenosine tri-phosphate (ATP) synthase β, enolase α, and several other unannotated proteins. A major limitation of this study was the limited use of proteomics to identify a very small number of autoantigens and the use of a very limited sample size. However, this study suggests that several autoantigens like tTG exist in patients with CeD against which serum antibodies may be developed.

### 5.3. Metabolomics Approaches

While there is a relative paucity of proteome-based studies, many metabolomic studies have been done in patients with CeD to find metabolic markers of CeD, presumably due to the fact that metabolites can be detected with greater sensitivity and with much lower requirement of biological material. Typically, mass spectrometric (MS) and nuclear magnetic resonance (NMR) methods are used for metabolomics [93]. Nuclear Magnetic Resonance studies that have been carried out with urine and sera of patients with CeD in comparison to those of healthy controls have revealed several metabolic differences between the two study groups [94]. Sera of CeD patients have demonstrated lower levels of amino acids, lipids, pyruvate, and choline and higher levels of glucose and 3-hydroxybutyric acid in comparison to the healthy controls. Differentiation of patients with CeD from healthy controls was done by ^1^H NMR studies using serum and urine metabolomics, where a pattern of metabolites was observed [95].

Furthermore, using partial least squares-discriminant analysis of metabolites expressed in the small intestinal mucosa, we observed a clear distinction in the pattern of metabolites in the intestinal mucosa of patients with CeD and controls. There was a significantly higher concentration of isoleucine, leucine, aspartate, succinate, and pyruvate, and a lower concentration of glycerol-phosphocholine in the duodenal mucosa of patients with CeD patients compared with controls, suggesting abnormalities in glycolysis, Krebs cycle, and amino acid metabolism in patients with CeD.

While studies based on metabolomic approach have provided a distinctive pattern of metabolites which can differentiate patients with CeD from controls, such a pattern is based on a large panel of metabolites, rather than based on a few metabolites. Hence, it is less likely that such a pattern of metabolites can be tested individually in a clinical laboratory.

## 6. Biomarker to Predict Dietary Adherence

The mainstay of treatment of CeD is a life-long GFD with strict adherence. It is hard for the patient to achieve the high adherence rate because of widespread use of gluten in the food industry [96]. Approximately one-third of patients with CeD are not able to comply with GFD very well and even those who do, a persistence of enteropathy is observed in 20–40% of patients, which most likely attributable to inadvertent use of gluten by cross contamination [97,98]. It is therefore important to monitor the level of adherence to GFD by a biomarker which can detect ingestion of gluten by patients with CeD.

### 6.1. Gluten Immunogenic Peptide

Gluten immunogenic peptide (GIP) are fragments of gluten which are resistant to digestion and therefore eliminated in the urine and the stool [99]. The presence of gluten in the urine or stool indicates recent consumption of gluten-containing food [100]. The α-gliadin 33-mer is the main immunodominant toxic peptide that interacts with the immune system of patients with CeD and a proportion of this peptide is absorbed in the GI tract and makes it way from blood to and partly excreted in the stool [99].

The gluten peptide fraction has been detected in the stool of healthy subjects after consumption of not only a normal gluten-containing diet but even in those who have consumed even <100 mg gluten/day. The level of gluten peptides detected in the stool has been shown to be in proportionate with the amount of gluten ingestion [101]. Gluten Immunogenic Peptide is also detectable in urine, and a positive correlation was observed with the amount of gluten intake and amount excreted in the urine. Gluten Immunogenic Peptide can be detected in the urine even with 25 mg of gluten ingestion.

Gluten Immunogenic Peptide is detected by anti-GIP immunochromatographic strips and the conduct of the test is very simple and the test has high sensitivity and specificity [102]. The GIP detection kit is similar to the pregnancy test, where GIP present in the sample, react with specific antibody on the strips. As GIP is rapidly cleared through urine, the presence of GIP in the urine suggests the ingestion of gluten within 20 h. Since the GIP stays longer in the intestinal lumen before getting completely excreted, the detection of GIP in stool suggests the ingestion of gluten within 2–7 days. Approximately 17–80% of patients with CeD who are following GFD have been found to have GIP in their stool [102].

### 6.2. Mean Platelet Volume

Mean platelet volume (MPV) has been recognized as a marker of inflammation in various diseases such as ulcerative colitis, acute pancreatitis and myocardial infraction etc. [103,104,105]. Purnak et al. have shown significantly higher MPV in CeD compared to healthy subjects and significant decrease in MPV in those showing good compliance to GFD compared to non-compliant individuals [106].

## 7. Miscellaneous Biomarkers

### 7.1. MicroRNAs

MicroRNAs (miRNAs) are small, endogenous noncoding RNAs that act as post-transcriptional regulators of gene expression. Upon cell death, miRNAs are released into the surrounding environment and then reach peripheral blood circulation or body fluids, hence detection of tissue- specific or tissue-enriched miRNA in biofluids might be used as a biomarker for specific tissue damage and specific tissue event. In a study, including untreated adult CeD patients, treated CeD patients and control subjects without CeD showed dysregulation of seven miRNAs such as miR-31-5p, miR-192-3p, miR-194-5p, miR-551a, miR-551b-5p, miR-638 and miR-1290 in patients with CeD compared to those without CeD [107]. In another study, the expression pattern of miR-21 and miR-31 was assessed in pediatric patients with CeD using qRT-PCR where significant up-regulation of miR-21 and down-regulation of miR-31 was observed in the untreated celiac patients in comparison with the treated group (n = 25) and healthy controls (n = 20). Furthermore, there was a correlation between the expression of miR-21 and the titer of anti- tTG Ab [108]. Many miRNAs including specifically miR-21 and miR-31 require further exploration.

### 7.2. Biomarkers for Assessment of Intestinal Permeability

Assessment of intestinal permeability is performed to assess the overall function of transport through the intestinal epithelial paracellular route [109]. Urinary excretion of disaccharides and monosaccharides and the ratio of their excretion is a basis for the measurement of intestinal permeability. Intestinal permeability can be assessed using a variety of marker probes such as lactulose, mannitol, rhamnose and cellobiose, polyethylene glycol (PEG) 400, PEG 1000, ^51^Cr-EDTA and ^99m^Tc-DTPA. The lactulose and mannitol ratio is the most commonly used test for assessment of small intestinal permeability. The majority of treatment naïve patients with CeD have abnormal intestinal permeability [109]. While the withdrawal of gluten improves both clinical and histological abnormalities, it also normalizes the paracellular function and hence the intestinal permeability in them.

## 8. Conclusions

While many of the currently used biomarkers for the screening and diagnosis of CeD are quite reliable, there is need for blood biomarkers which can predict presence of villous abnormalities even without performing intestinal mucosal biopsies and biomarkers which can assist in the monitoring of the disease. While initial data on the performance of biomarkers such as plasma citrulline and plasma I-FABP in the prediction of the presence of villous abnormalities even without obtaining intestinal mucosal biopsies is very promising, there is a need for further validation of these biomarkers before they can be used reliably in the clinical practice. GIP detection in the stool or in the urine is now used in the clinical practice for detection of gluten ingestion; a positive test however, only reflects ingestion of gluten over the past few days, before the performance of the test. What really is required is a test which can tell us the adherence to gluten ingestion over a longer period of time. In order to understand the fundamental biology and in search of the clinically usable biomarker, a number of research laboratories are exploring various approaches such as transcriptomics, proteomics, and genetics, and we hope that these powerful technologies will provide us in the near future with clinically usable biomarkers for the diagnosis, treatment, and monitoring of the diseases.

## Figures and Tables

**Table 1 jcm-08-00885-t001:** Established and potential biomarkers for celiac disease

Types of Biomarkers	Name of the Biomarker
Serological biomarker	Anti-gliadin antibody (AGA)
	Anti-endomysial antibody (AEA)
	Anti-tissue transglutaminase (tTG)- TG2, TG6, TG3
	Deamidated gliadin peptide (DGP)
	Synthetic neo-epitopes tTG-DGP complex
Genetic marker	HLA-DQ haplotyping
Biomarkers for the prediction of enteropathy	Cytochrome P450 3A4 (CYP3A4)
	Plasma citrulline
	Intestinal-fatty acid binding proteins
	Regenerating gene1α (Reg 1α)
Biomarkers to predict dietary adherence	Gluten immunogenic peptide (GIP)
	Mean platelet volume (MPV)
Miscellaneous Biomarkers	miRNA Intestinal permeability

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
