# Peer review of "Non-Invasive Biomarkers for Celiac Disease"

_jcm, 2019, doi:10.3390/jcm8060885_

Reviewer 1 Report

 Reviewer’s Comments to Authors:

 The manuscript (MS) titled ‘Non-Invasive Biomarkers for Celiac Disease’ by Alka Singh et.al. has been fairly well written, but can be improved substantially.  The authors’ attempt at presenting a comprehensive review fell short on the novelty front, and does not add much to the existing literature on the diagnosis of Celiac Disease (CD).  Overall, this MS needs a significant improvement and changes to general writing style, to better engage the readers of the Journal of Clinical Medicine (JCM). 

 The authors need to clarify the aspects listed below to be acceptable for publication. 

 1. Line 8:  While listing a wide range (37 – 59 million) for CD sufferers, there is a discrepancy in numbers in the Introduction (line 34).  This can be presented as a % or refine the range with consistency in the MS.

 2. Line 13:  what is 'celiac-specific antibody'? Please clarify.

 3. Lines 18 to 26 from Abstract were just copied and pasted in Introduction (lines 41 to 49).  The Introduction can be substantially improved by adding material about the clinical manifestations of CD, as this is an MS representing CD clinical aspects.    

 4.  In Table 1? (only table in the MS!), how is POCT (a test) a serological biomarker?  Also, how is High-throughput Approach (Proteomics etc.) a biomarker?  Please explain.

 5. Lines 89-90: ‘high rate of inter- and intra-observer variations…’  What does this mean? Is it analysts? Please explain.

 6. It is nice that the authors included a gist of their research in the MS.  But, please expand on the results/observations for making this Review more comprehensive for the JCM readers.  Considering the high number of References listed, the MS does not capture their effect in becoming/seeming comprehensive!!

 7. Expand on the Conclusions section, as the current material is not sufficient.  

 8. Although a minor aspect, I encourage the authors to proof-read the MS carefully for grammatical corrections and consistency (e.g. line 18 & line 41 ‘they’ for ‘there’; line 15- abbreviation CeD introduced abruptly etc.).

Author Response

Our point to point responses 

Reviewer #1 The authors need to clarify the aspects listed below to be acceptable for publication

Comments to the Author

Comment #1. Line 8:  While listing a wide range (37 – 59 million) for CD sufferers, there is a discrepancy in numbers in the Introduction (line 34).  This can be presented as a % or refine the range with consistency in the MS.

Response:  Our apology for the discrepancy in the data. Based on the global prevalence of biopsy confirmed CeD of 0.68% (95% CI 0.5%, 0.9%) and global population of 6.4 billion; the expected number of people having CeD is 37-59 million, globally.

Comment #2.    Line 13:  what is 'celiac-specific antibody'? Please clarify. 

Response: We appreciate the comments of the reviewer. The celiac-specific serological tests include anti-tissue transglutaminase antibody (anti-tTG Ab), anti-endomysial antibody (EMA) and anti-deamidated gliadin peptide (anti-DGP Ab). We have clarified this in the revised manuscript. 

Comment #3.    Lines 18 to 26 from Abstract were just copied and pasted in Introduction (lines 41 to 49).  The Introduction can be substantially improved by adding material about the clinical manifestations of CD, as this is an MS representing CD clinical aspects.     

Response: Thanks for the observation. We have modified the introduction accordingly and we have added clinical manifestations of CeD in the introduction segment. 

Comment #4.    In Table 1? (only table in the MS!), how is POCT (a test) a serological biomarker?  Also, how is High-throughput Approach (Proteomics etc.) a biomarker?  Please explain.

Response: We have made appropriate changes, as suggested. 

Comment #5.    Lines 89-90: ‘high rate of inter- and intra-observer variations…’  What does this mean? Is it analysts? Please explain.

Response: We have deleted this line.

Comment #6.    It is nice that the authors included a gist of their research in the MS.  But, please expand on the results/observations for making this Review more comprehensive for the JCM readers.  Considering the high number of References listed, the MS does not capture their effect in becoming/seeming comprehensive!!

Response: Thanks for the observations. We have expanded the text at appropriate places. 

 Comment #7.    Expand on the Conclusions section, as the current material is not sufficient.  

Response: Thank you so very much, we have expanded the conclusion segment. 

Comment #8.    Although a minor aspect, I encourage the authors to proof-read the MS carefully for grammatical corrections and consistency (e.g. line 18 & line 41 ‘they’ for ‘there’; line 15- abbreviation CeD introduced abruptly etc.).

Response: Manythanks for the observation, we have read the manuscript again and made necessary changes. 

Reviewer 2 Report

This is a solid and complete review about biomarkers in celiac disease.

I have not major methodological concerns.

Minor points: page 5, illous instead of villous (typing error). 

In the same section (markers to predict...) the authors affirm that "periodic biopsies are also recommended for the monitoring of the disease". I do not agree with this sentence. According to common guidelines, periodic biopsies usually are not necessary for the monitoring of celiac disease, except in case of complicated or refractory to the gluten free diet disease.

Author Response

Our point to point responses 

Reviewer #2 Comments and suggestions for the Authors

This is a solid and complete review about biomarkers in celiac disease.

I have not major methodological concerns.

Comment #1. Page 5, illous instead of villous (typing error)

Our response: Our apology, we have corrected the error. 

Comment #2. Page 5, in this section (markers to predict…..) the authors affirm that “periodic biopsies are also recommended for the monitoring of the disease”. I do not agree with this sentence. According to common guidelines, periodic biopsies usually are not necessary for the monitoring of celiac disease except in case of complicated or refractory to the gluten free diet disease.

Response: We do agree with the reviewer’s comment that periodic biopsies are not recommended for monitoring of disease at this point of time.  However, one of the goals of treatment of celiac disease is mucosal healing. The rate of complete mucosal recovery (Modified Marsh grade 0) is not very high and 30-50% of patients persist to have some form of enteropathy (modified Marsh grade 1 to grade 3) even after one to five years of gluten-free diet. Persistent enteropathy is considered to be due to advertent or inadvertent use of gluten in the diet. 

We have deleted this part in the revised manuscript. 
